# When Federated Recommendation Meets Cold-Start Problem: Separating Item Attributes and User Interactions

## ABSTRACT

Federated recommendation systems usually trains a global model on the server without direct access to users' private data on their own devices. However, this separation of the recommendation model and users' private data poses a challenge in providing quality service, particularly when it comes to new items, namely cold-start recommendations in federated settings. This paper introduces a novel method called Item-aligned Federated Aggregation (IFedRec) to address this challenge. It is the first research work in federated recommendation to specifically study the cold-start scenario. The proposed method learns two sets of item representations by leveraging item attributes and interaction records simultaneously. Additionally, an item representation alignment mechanism is designed to align two item representations and learn the meta attribute network at the server within a federated learning framework. Experiments on four benchmark datasets demonstrate IFedRec's superior performance for cold-start scenarios. Furthermore, we also verify IFedRec owns good robustness when the system faces limited client participation and noise injection, which brings promising practical application potential in privacy-protection enhanced federated recommendation systems. The implementation code is available[1].

**Relevance Statement:** This paper aims to address the challenge of conducting cold-start items recommendation to users in federated recommendations, which aligns with the scope of the user-modeling and recommendation track.

## CCS CONCEPTS

• **Information systems** → **Recommendation**; **User Modeling**.

## KEYWORDS

Federated Learning, Recommendation Systems, Cold-Start

**ACM Reference Format:**
Anonymous Author(s). 2018. When Federated Recommendation Meets Cold-Start Problem: Separating Item Attributes and User Interactions. In *Proceedings of International World Wide Web Conference (WWW'24)*. ACM, New York, NY, USA, 11 pages. https://doi.org/XXXXXXX.XXXXXXX

## 1 INTRODUCTION

---
[1]https://anonymous.4open.science/r/IFedRec-1257

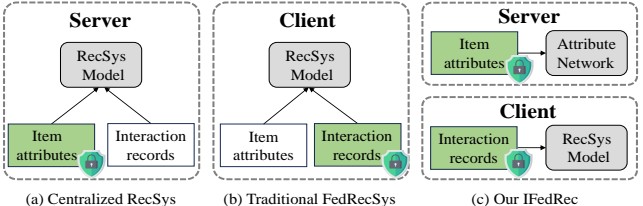

**Figure 1: Three cold-start recommendation systems comparison. The centralized method (a) saves raw item attributes on the server but exposes private user interaction records. Traditional FedRecSys (b) secures the interaction records but exposes the item attributes to the clients. Our IFedRec (c) can protect these two types of security-sensitive information.**

Cold-start is a long-standing challenge in the recommendation system research [36]. It demands the system's capability to infer recommendations for new items. To solve the cold-start issue, it is crucial to integrate the raw item attributes into the model for offering beneficial information, which have been verified as an effective scheme [7, 15, 33] in the centralized recommendation service setting. Generally, the service provider can collect all the users' personal data (*e.g.,* interaction records) and the items' raw attributes for model construction, as shown in Figure 1 (a). By learning the correlations between item attributes and user interaction records, the system can make predictions for the new items.

However, with the serious social concerns about the exploitation of user privacy [25, 30], developing recommendation models while protecting user's private data from being leaked has attracted increasing attention. As an emerging privacy-preserving recommendation framework, Federated Recommendation System (FedRecSys) [6, 22, 24] deploys individual models on the devices (clients), and a server can optimize a common model by commanding the local model parameter aggregation and distribution. Privacy can be guaranteed as users preserve private data locally, which prevents accessibility from the server or other users. Although impressive progress has been shown [34, 35], there is still a lack of solutions for cold-start recommendation models under the federated setting.

Given the remarkable success of cold-start recommendation models in the centralized setting, the intuitive idea to develop the federated version is to deploy the centralized model on each device, that is, each client downloads all the raw item attributes from the server and trains the local model with personal interaction records, as shown in Figure 1 (b). However, the dissemination of raw item attributes outside of the service provider poses a significant risk to the system. Firstly, the raw item attributes are crafted carefully with expert effort, and the disclosure can lead to substantial damage to commercial properties. Moreover, publicly available raw item attributes are susceptible to malicious usage and may incur hostile adversarial attacks [17, 37]. Hence, it is crucial to preserve the raw item attributes on the server. The challenge of constructing a cold-start FedRecSys lies in how to promote the system learning while

preserving the security of private interaction data on the client and the raw item attributes on the server.

In this paper, we present a novel **I**tem-aligned **Fed**erated aggregation framework for cold-start **Rec**ommendation (**IFedRec**), which is the first effort to achieve cold items recommendation in federated setting. To realize the cold-start FedRecSys while preserving user data and raw item attributes safely, we propose a coherent learning process for two item representations from the client and server. The client maintains item embeddings based on user interaction records capturing user preferences, while the server incorporates a meta attribute network to represent item attributes using raw item attributes. We also devise an item representation alignment mechanism to bridge the connection between item attributes and user preferences, enabling cold-start recommendation. Figure 1 (c) demonstrates how our IFedRec framework effectively executes cold-item recommendation by leveraging item attributes, while simultaneously ensuring the security of private interaction data and raw item attributes, preventing their exposure.

To implement the idea, we develop a two-phase learning framework, *i.e.,* learning on warm items and inference on cold items. In the learning phase, the server aggregates the local item embeddings to achieve the global one, which is then used as supervision to train the meta attribute network on the server. For each client, the local model training is carlibrated by minimizing the distance between local item embedding and item attribute representation from the server. This mechanism injects the attribute information into the recommendation model, which enhances item representation learning and promotes recommendation prediction. In the inference phase, the server learns the attribute representations for cold items. Each client can then utilized these attribute representations along with the user-specific recommendation models to make personalized recommendations. We integrate our framework into two representative FedRecSys, which gain significant performance improvement than the original version when dealing with cold-start scenarios. Our IFedRec achieves the state-of-the-art performance on four cold-start recommendation datasets, outperforming both federated and centralized baselines across comprehensive metrics. Moreover, we empirically demonstrate the robustness of IFedRec even when only a few clients participate in each communication round, which indicates its potential for practical application. Additionally, by integrating the local differential privacy technique, our IFedRec strikes a balance between model performance and system noise injection, which sheds lights on the privacy-protection enhanced FedRecSys construction.

In summary, our **main contributions** are listed as follows,

- We present a novel framework, IFedRec, to the best knowledge of the authors, it is the first effort to solve the cold-start recommendation under the federated setting where there are no interactions for the new items.
- Our method achieves state-of-the-art performance in extensive experiments and in-depth analysis supports the significance of cold items recommendation.
- The proposed item semantic alignment mechanism can be easily integrated into existing federated recommendation frameworks for cold-start recommendation performance improvement.

## 2 RELATED WORK

### 2.1 Cold-Start Recommendation

Cold-start recommendation research focuses on addressing the challenge of providing quality recommendation service for new items [38]. Several approaches have been proposed to tackle this issue, including collaborative filtering techniques [5, 32, 38], content-based methods [11, 27] and the hybrid models [4]. Collaborative filtering methods infer the item similarities based on historical user interactions and identify items that tend to be consumed together. Content-based methods leverage the item attributes to understand the item characteristics so that the system can analyze the correlations between new items and existing items and make recommendations. Hybrid models combine both collaborative filtering and content-based methods, which extract meaningful features from item attributes and integrate them into the collaborative filtering framework to discover the correlations with user interactions.

### 2.2 Federated Recommendation System

**Fed**erated **Rec**ommendation (**FedRec**) has recently drawn widespread attention due to the urgency of user privacy protection. Generally, each user is regarded as a client who trains a recommendation model with locally reserved private data, and a server coordinates the collaborative optimization among all clients by aggregating the model parameters. Various recommendation benchmark architectures have been adapted to the federated recommendation frameworks [3, 9, 10, 18, 20, 23, 35, 37]. However, existing FedRec models focus on recommending items with historical interactions, and the cold-start recommendation has rarely been studied. After thorough investigation, we found that only one FedRec model [28] is proposed for the item cold-start recommendation, which still depends on a small number of interactions of the new items. In this paper, we explore the setting that the system recommends the new items without any interactions, which is rather challenging and realistic in practical applications.

## 3 PRELIMINARY

**Federated Cold-Start Recommendation.** Let $\mathcal{U}$ denote the user set with $n = |\mathcal{U}|$ users. $\mathcal{I}^{warm}$ represents the warm item set that has been interacted by users, and $\mathcal{I}^{cold}$ is the cold item set whose items have never been interacted by any users. Given the item attribute matrix $\mathcal{X}^{warm}$ and each user's interaction records $\mathcal{Y}_u^{warm}$, federated cold-start recommendation aims to a recommendation model $\mathcal{F}_\theta$, so that the system can make recommendations for each user about the cold items based on the item attribute matrix $\mathcal{X}^{cold}$. Mathematically, the prediction could be formulated as follows,

$$\mathcal{Y}_u^{cold} = \mathcal{F}_\theta(\mathcal{X}^{cold}) \tag{1}$$

Particularly, $\mathcal{F}_\theta$ consists of three modules, *i.e.,* an item embedding module $\mathcal{P}$, a user embedding module $\mathcal{Q}$ and a rating prediction module $\mathcal{S}$, and let $\theta := (p, q, s)$ denote the model parameters.

## 4 METHODOLOGY

In this section, we begin by introducing the overall framework of the proposed method. We then delve into the details of the learning phase workflow and summarize it as an optimization algorithm.

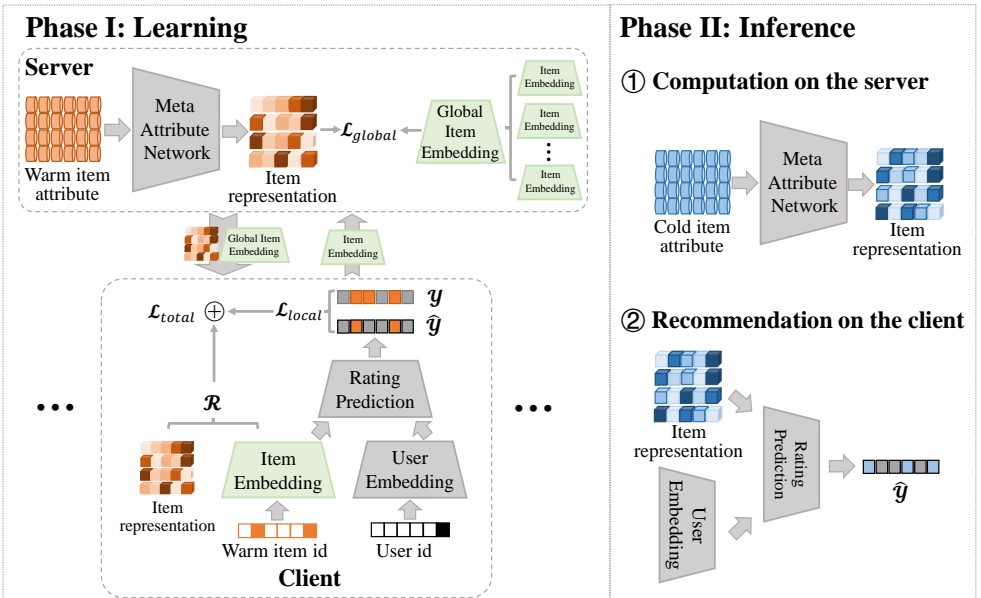

**Figure 2: The framework of IFedRec. During the *learning* phase, the client uploads the item embedding to the server for global aggregation, and other recommendation modules are preserved locally to capture user personalization. On the server side, we elaborate a meta attribute network to learn item attribute representation based on raw item attributes. Besides, an item representation alignment mechanism is developed to align two item representations, *i.e.,* $\mathcal{L}_{global}$ and $\mathcal{R}$. During the inference phase, the server first learns the cold item attribute representation, and then each client can make personalized recommendations by integrating it with locally preserved recommendation modules.**

Furthermore, we demonstrate the application of the inference phase specifically for recommending cold items. Finally, we present our IFedRec that enhances privacy protection by incorporating the local Differential Privacy technique.

## 4.1 Framework Overview

Introducing raw item attributes is crucial to achieve cold-start recommendation. However, simply utilizing the raw item attributes to learn item embeddings may risk commercial property damage and lead to adversarial attack to the FedRecSys. In this context, we develop a novel **I**tem-aligned **Fed**erated aggregation for cold-start **Rec**ommendation (**IFedRec**) model, whose overall framework is illustrated in Figure 2. We elaborate two phases to firstly model the item information and then utilize the trained model to infer the cold start items. **During the learning phase**, each client trains a recommendation model locally, and the server learns a meta attribute network globally. We present an item representation alignment mechanism to align two item representations, so that the system can learn enhanced item representation and achieve cold-start recommendation. **During the inference phase**, the server first learns the cold item attribute representation, and then each user can make a prediction using it with the help of locally preserved personalized recommendation modules.

## 4.2 Learning on the Warm Items

To achieve a model that can make recommendations on new items, we first train the model on the warm items based on the user interaction records and raw item attributes. To be specific, we alternately perform the following two steps: **First**, the server trains the global meta attribute network $\mathcal{M}_{\phi}$ with the item attributes.

**Second**, each client $u$ updates the local recommendation model $\mathcal{F}_{\theta_u}$ with the historical interaction records. **Meanwhile**, an item representation alignment mechanism is introduced to align item attribute representation from the server and item embedding from the client. Next we detail the two steps below.

*4.2.1 Global meta attribute network learning.* Under the federated learning optimization framework, the server is responsible for coordinating all clients to train a globally shared model. In our method, we regard the item embedding module $\mathcal{P}$ as the shared component, which is learned from user interactions. Both user embedding and rating prediction modules are regarded as private components and preserved locally. Once the clients have completed the local model training, they upload the item embeddings to server. Then, the server aggregates all received item embeddings into a global one, which depicts the common item characteristics derived from user preferences. Particularly, we adopt the naive average aggregation formulation due to its simplicity and no additional computational overhead, which is as follows,

$$p := \frac{1}{n} \sum_{i=1}^{n} p_u \qquad (2)$$

where $p_u$ denotes the item embedding parameter of client $u$ and $n$ is the total number of clients. Other weight-based aggregation methods [16, 19] are also promising for better performance. After aggregation, the global item embedding would be distributed to clients so that the common item characteristics can be exchanged among clients.

Generally, the server holds rich attributes of items, including both warm items and cold items. The item attribute information

can be used to bridge the connection between items, which paves the way to cold item recommendation. Specifically, we propose a meta attribute network $\mathcal{M}_\phi$ to learn the item representation based on item attributes and deploy it on the server. Compared with the on-device deployment, we preserve the raw item attributes on the service provider, which guarantees the data safety from exposure and alleviates the potential damage of malicious utilization. Particularly, We formulate the learning of $\mathcal{M}_\phi$ as,

$$r_v := \mathcal{M}_\phi(x_v) \tag{3}$$

where $\phi$ is the model parameter. The $x_v$ and $r_v$ are the attribute and learned representation of item $v$, respectively.

**Item embedding alignment.** We regard the global item embedding $p$ as the supervision to train the meta attribute network $\mathcal{M}_\phi$, so that we can construct the connection between the item attributes and the user interaction records with item embedding as the intermediary. Then, for the cold items, which have only attribute information, our method can calculate the attribute representation and make recommendations for them. Particularly, considering the properties of the regression task, we adopt the *mean square error* as the loss function and formulate it as,

$$\mathcal{L}(p;\phi) := \frac{1}{m} \sum_{v=1}^{m} (r_v - p(v))^2 \tag{4}$$

where $m$ is the number of warm items. $r_v$ and $p(v)$ are the learned attribute representation and global item embedding of item $v$.

Based on the loss $\mathcal{L}$ in Eq. (4), we update the meta attribute network parameter $\phi$ via stochastic gradient descent algorithm and the $t$-th update step is,

$$\phi^t := \phi^{t-1} - \gamma \partial_{\phi^{t-1}} \mathcal{L}(p;\phi) \tag{5}$$

where $\gamma$ is the parameter update learning rate.

*4.2.2 Local recommendation model update.* Based on the recommendation model $\mathcal{F}_\theta$, where $\theta := (p, q, s)$, we formulate the model prediction of user $u$ about item $v$ as,

$$\widehat{\mathcal{Y}}_{uv} := \mathcal{S}(\mathcal{P}_v, \mathcal{Q}_u) \tag{6}$$

where $\mathcal{P}_v$ and $\mathcal{Q}_u$ denote the embedding of item $v$ and user $u$, respectively. Particularly, we discuss the typical implicit feedback recommendation task, *i.e.*, $\mathcal{Y}_{uv} = 1$ if there is an interaction between user $u$ and item $v$; otherwise $\mathcal{Y}_{uv} = 0$. With the binary-value nature of implicit feedback, we define the recommendation loss of user $u$ as the *binary cross-entropy loss*,

$$\mathcal{L}_u(\mathcal{Y}_{uv};\theta_u) := - \sum_{(u,v) \in D_u} \log \hat{y}_{uv} - \sum_{(u,v') \in D_u^-} \log(1 - \hat{y}_{uv'}) \tag{7}$$

where $\mathcal{D}_u^-$ is the negative samples set of user $u$. It is worth noting that other loss metrics can also be adopted, and here we take the binary cross-entropy loss as an example. To construct $\mathcal{D}_u^-$ conveniently, we first count all the uninteracted items of user $u$ as,

$$\mathcal{I}_u^- := \mathcal{I}^{warm} \backslash \mathcal{I}_u \tag{8}$$

where $\mathcal{I}_u$ is the interacted warm items set of user $u$. Then, we uniformly sample negative items from $\mathcal{I}_u^-$ by setting the sampling ratio based on the user's interacted item amount.

**Item attribute representation alignment.** For the local recommendation model, it learns a unique item embedding for each item, which depicts the item characteristic. Meanwhile, the server can learn the latent representation based on the raw item attribute,

which is effective complementary information that can be further used to enhance client model training leading to a more comprehensive local item embedding.

To this end, we propose to align the local item embedding module with the global learned item attribute representation. Particularly, we regard the item attribute representation as a regularization term to enrich the recommendation model supervision information, and reformulate the local model training loss as,

$$\mathcal{L}_{total} := \mathcal{L}_u(\mathcal{Y}_{uv};\theta_u) + \lambda \mathcal{R}(p_u, r) \tag{9}$$

where $p_u$ is the item embedding module parameter of user $u$ and $r$ denotes the item attribute representation learned by raw item attributes on the server side.

Based on the local training loss $\mathcal{L}_{total}$, we can update the recommendation model parameter $\theta_u$ via stochastic gradient descent algorithm. Notably, we adopt the alternative update method to update different modules, *i.e.*, first update the locally preserved user embedding module $\mathcal{Q}$ and rating prediction module $\mathcal{S}$ to adapt the recommendation model with the global item embedding, and then update the local item embedding $\mathcal{P}$ with the tuned $\mathcal{Q}$ and $\mathcal{S}$. The $t$-th update step is formulated as,

$$\begin{aligned} (q_u^t, s_u^t) &:= (q_u^{t-1}, s_u^{t-1}) - \eta_1 \partial_{(q_u^{t-1}, s_u^{t-1})} \mathcal{L}_{total} \\ p_u^t &:= p_u^{t-1} - \eta_2 \partial_{p_u^{t-1}} \mathcal{L}_{total} \end{aligned} \tag{10}$$

where $\eta_1$ and $\eta_2$ are the parameter update learning rate for modules $\mathcal{Q}$ and $\mathcal{S}$, and module $\mathcal{P}$, respectively.

*4.2.3 Overall optimization objective.* Under the FL setting, we regard each user as a client $u$, who trains a local recommendation model $\mathcal{F}_\theta$ based on private dataset $D_u$. To sum up, we formulate the proposed IFedRec as the below bi-level optimization problem,

$$\min_{\{p_u,q_u,s_u\}_{u=1}^n} \sum_{u=1}^{n} \mathcal{L}_u(\mathcal{Y}_u; p_u, q_u, s_u) + \lambda \mathcal{R}(p_u, r) \tag{11}$$
$$s.t. \qquad r := \mathcal{M}_\phi(\mathcal{X}^{warm})$$

where $n$ is the number of clients. The $\mathcal{L}_u$ is the supervised loss of the $u$-th client. $p_u$, $q_u$, and $s_u$ are item embedding, user embedding, and rating prediction module parameters, respectively. $\mathcal{R}(\cdot, \cdot)$ is the regularization term and $\lambda$ is the regularization coefficient. The $r$ is learned by the meta attribute network $\mathcal{M}_\phi$ with item attributes $\mathcal{X}^{warm}$ as input. Particularly, we aggregate the clients' item embeddings to achieve global item embedding $p$ and take it as the supervision to optimize the meta attribute network on the server.

*4.2.4 Algorithm.* We summarize the optimization procedure on the warm items set into Algorithm 1. The optimization objective can be solved with multiple communication rounds between the server and clients. Initially, the server initializes the global shared item embedding module parameter $p$ and the meta attribute network parameter $\phi$ (*lines 1-2*). In each round, the server first updates the meta attribute network $\mathcal{M}_\phi$ with the global item embedding $p$ as supervision, and then compute the item attributes representation $r$ with the updated $\mathcal{M}_\phi$ (*lines 4-8*). Then, the server samples some clients to participate in the current round's training and aggregates the global item embedding $p$ with received updated ones from clients (*lines 9-13*). For each client, it first initializes the recommendation model $\mathcal{F}_\theta$ with the distributed global item embedding and the latest user embedding and rating prediction parameter (*lines*

**Algorithm 1** Item-Guided Federated Aggregation for Cold-Start Recommendation - **Learning on the Warm Items**

**ServerExecute:**
1: Initialize item embedding module parameter
2: Initialize meta attribute network parameter
3: **for** each round $t = 1, 2, ...$ **do**
4:     **for** $e$ from 1 to $E_1$ **do**
5:         Compute $\mathcal{L}(p^t; \phi)$ with Eq. (4)
6:         Update $\phi^t$ with Eq. (5)
7:     **end for**
8:     Compute warm items representation $r_{warm}^t$ with Eq. (3)
9:     $S^t \leftarrow$ (select a client subset randomly from all $n$ clients with sampling ratio $\alpha$)
10:     **for** each client $u \in S^t$ **in parallel do**
11:         $p_u^{t+1} \leftarrow$ ClientUpdate$(t, u, p^t, r_{warm}^t)$
12:     **end for**
13:     Aggregate global item embedding $p^{t+1}$ with Eq. (2)
14: **end for**

**ClientUpdate$(t, u, p, r)$:**
1: Initialize $p_u$ with $p$
2: **if** $t = 1$ **then**
3:     Initialize user embedding module parameter $q_u$
4:     Initialize rating prediction module parameter $s_u$
5: **else**
6:     Initialize $q_u$ and $s_u$ with the latest updates
7: Count all uninteracted items set $\mathcal{I}_u^-$ with Eq. (8)
8: Sample negative feedback $D_u^-$ from $\mathcal{I}_i^-$
9: $\mathcal{B} \leftarrow$ (split $D_u \cup D_u^-$ into batches of size $B$)
10: **for** $e$ from 1 to $E_2$ **do**
11:     **for** batch $b \in \mathcal{B}$ **do**
12:         Compute $L_{total}$ with Eq. (9)
13:         Update $(p_u, q_u, s_u)$ with Eq. (10)
14:     **end for**
15: **end for**
16: **Return** $p_u$ to server

1-6). Then, the client prepares the local training dataset by sampling negative items and divides it into batches (*lines 7-9*). Finally, the client updates the recommendation model and uploads the latest item embedding to the server (*lines 10-16*).

### 4.3 Inference on the Cold Items

During the learning phase, the system is optimized with the warm items and the learned model can be used for inferring cold item recommendations. When new items $\mathcal{I}^{cold}$ come, the server first calculates the item representation $r_{cold}$ via the meta attribute network. Then, the clients can combine $r_{cold}$ with the locally preserved user embedding $Q$ and rating prediction module $S$ to make personalized recommendations. We summarize the procedure in Algorithm 2.

### 4.4 Privacy-Protection Enhanced IFedRec with Local Differential Privacy

To further enhance the privacy-protection, we can integrate privacy-preserving techniques into FL optimization framework, such as Differential Privacy [8] and Homomorphic Encryption [2], and the

**Algorithm 2** Item-Guided Federated Aggregation for Cold-Start Recommendation - **Inference on the Cold Items**
1: Compute cold items representation $r_{cold}$ with Eq. (3)
2: **for** each client **in parallel do**
3:     Construct recommendation model with the latest $Q$ and $S$
4:     Assign $\mathcal{P}$ with $r_{cold}$
5:     Make cold items recommendation with Eq. (2)
6: **end for**

key idea is to prevent the server from inferring the client private information through the received model parameters. In our method, each client uploads the item embedding module to the server for exchanging common information, which may be maliciously used to infer sensitive user information. To handle the issue, we present a privacy-protection enhanced IFedRec by equipping it with the local Differential Privacy technique. Particularly, each client $u$ adds a zero-mean Laplacian noise to the item embedding before uploaded to the server, which can be formulated as,

$$p_u = p_u + Laplace(0, \delta) \tag{12}$$

where $\delta$ is the noise strength. As a result, the server receives an encrypted item embedding from clients, which reduces the risk of user privacy exposure.

## 5 EXPERIMENT

In this section, we conduct experiments to evaluate our method and explore the following research questions:
**Q1**: How does IFedRec perform compared with the federated models and the state-of-the-art centralized models?
**Q2**: Why does IFedRec work well on cold-item recommendation?
**Q3**: How do the key hyper-parameters impact the performance?
**Q4**: How well does IFedRec converge w.r.t. the client's amount?
**Q5**: How does IFedRec perform under noise injection?

### 5.1 Datasets

We evaluate the proposed IFedRec on two cold-start recommendation benchmark datasets, *i.e.*, CiteULike [29] and XING [1], which have rich item attribute information. Particularly, we extract three dataset subsets from the original XING dataset according to user amount and mark them as XING-5000, XING-10000 and XING-20000, respectively. For a fair comparison, we follow the warm items and cold items division of [39]. For CiteULike, we select 80% items as the warm items, which serve as the training set to learn the model, and keep the other items as cold items. Then, we sample 30% items from the cold items as the validation set and take the remaining cold items as the test set. For three XING datasets, we divide the training set (warm items), validation set and test set (cold items) according to the ratio of 6:1:3. The dataset statistics and more descriptions about the datasets can be found in **Appendix A**.

### 5.2 Experimental Setup

***Evaluation metrics.*** We adopt three ranking metrics to evaluate model performance, *i.e.*, *Precision@k*, *Recall@k* and *NDCG@k*, which are common used evaluation metrics [7, 13, 39]. Particularly, we report results of $k = \{20, 50, 100\}$ in this paper.

| Methods | Metrics | CiteULike | | | XING-5000 | | | XING-10000 | | | XING-20000 | | |
|---|---|---|---|---|---|---|---|---|---|---|---|---|---|
| | | @20 | @50 | @100 | @20 | @50 | @100 | @20 | @50 | @100 | @20 | @50 | @100 |
| **FedRec** FedMVMF | Recall | 6.18 | 14.34 | 24.97 | 1.96 | 2.70 | 3.81 | 0.96 | 2.61 | 4.50 | 0.77 | 2.42 | 4.59 |
| | Precision | 1.57 | 1.48 | 1.30 | 0.77 | 0.46 | 0.36 | 0.52 | 0.55 | 0.48 | 0.51 | 0.66 | 0.62 |
| | NDCG | 5.55 | 10.04 | 14.42 | 2.60 | 1.78 | 1.98 | 1.02 | 1.85 | 2.43 | 0.65 | 1.50 | 2.39 |
| CS_FedNCF | Recall | 1.49 | 3.83 | 7.21 | 0.22 | 2.37 | 3.15 | 0.44 | 0.77 | 1.51 | 0.16 | 1.21 | 1.72 |
| | Precision | 0.37 | 0.39 | 0.36 | 0.14 | 0.41 | 0.29 | 0.24 | 0.17 | 0.17 | 0.10 | 0.33 | 0.23 |
| | NDCG | 1.76 | 3.16 | 4.38 | 0.26 | 0.96 | 1.20 | 0.37 | 0.42 | 0.67 | 0.16 | 0.67 | 0.93 |
| CS_PFedRec | Recall | 1.37 | 2.66 | 4.67 | 0.13 | 0.54 | 1.54 | 0.29 | 2.10 | 2.42 | 0.16 | 1.21 | 1.72 |
| | Precision | 0.33 | 0.25 | 0.24 | 0.09 | 0.13 | 0.18 | 0.19 | 0.44 | 0.26 | 0.19 | 0.33 | 0.23 |
| | NDCG | 1.40 | 1.92 | 2.54 | 0.15 | 0.34 | 0.93 | 0.35 | 1.02 | 0.99 | 0.16 | 0.67 | 0.93 |
| FedVBPR | Recall | 18.73 | 29.88 | 39.55 | 2.03 | 3.02 | 3.63 | 0.42 | 0.82 | 1.26 | 0.40 | 1.35 | 1.86 |
| | Precision | 3.75 | 2.46 | 1.66 | 0.78 | 0.56 | 0.36 | 0.24 | 0.19 | 0.14 | 0.27 | 0.36 | 0.24 |
| | NDCG | 13.24 | 16.07 | 17.91 | 0.95 | 1.37 | 1.41 | 0.35 | 0.48 | 0.57 | 0.32 | 0.74 | 0.98 |
| FedDCN | Recall | 1.42 | 3.57 | 6.59 | 0.32 | 0.65 | 1.14 | 0.43 | 0.83 | 1.52 | 0.24 | 0.80 | 1.43 |
| | Precision | 0.35 | 0.38 | 0.35 | 0.17 | 0.15 | 0.13 | 0.22 | 0.19 | 0.17 | 0.14 | 0.18 | 0.16 |
| | NDCG | 1.10 | 2.44 | 3.60 | 0.27 | 0.46 | 0.66 | 0.51 | 0.46 | 0.66 | 0.21 | 0.46 | 0.64 |
| **Ours** IFedNCF | Recall | **42.32** | **59.92** | **72.89** | **23.48** | **42.05** | **55.45** | **26.97** | **41.57** | **55.37** | **26.36** | **41.44** | **54.48** |
| | Precision | **9.70** | 5.80 | 3.65 | **13.66** | **9.55** | **6.37** | **14.38** | **9.02** | **6.06** | **16.25** | **10.23** | **6.75** |
| | NDCG | **34.29** | 37.61 | 38.74 | **20.93** | **27.41** | **29.46** | **21.65** | **24.66** | **27.02** | **21.99** | **25.30** | **27.22** |
| IPFedRec | Recall | 41.51 | 59.63 | 72.71 | 21.77 | 37.30 | 53.18 | 25.92 | 40.33 | 54.64 | 24.67 | 40.07 | 53.58 |
| | Precision | 9.48 | **5.81** | **3.67** | 12.75 | 8.76 | 6.12 | 13.84 | 8.77 | 5.97 | 15.29 | 9.92 | 6.66 |
| | NDCG | 33.48 | **37.69** | **39.07** | 19.74 | 24.77 | 28.34 | 20.66 | 23.90 | 26.52 | 20.53 | 24.49 | 26.91 |

**Table 1: Experimental results of the federated baselines and our method on four datasets. "FedRec" denotes the federated baselines and "Ours" represents that we integrate two state-of-the-art federated models into our framework. Particularly, we report the results in units of 1e-2 and the best results are bold.**

*Baselines.* We consider two branches of baselines: federated cold-start recommendation methods and centralized cold-start recommendation methods. For federated methods, we compare with the federated multi-view matrix factorization framework FedMVMF [10], and we adapt two state-of-the-art FedRec models [22, 35] into the cold-start setting (CS_FedNCF and CS_PFedRec). Besides, we choose two representative content enhanced centralized recommendation model, *i.e.,* VBPR [12] and DCN [31], and construct the federated version (FedVBPR and FedDCN) for a more comprehensive comparison. For centralized methods, we survey the recent cold-start recommendation papers and take two latest models (Heater [39] and GAR [7]) as our baselines. Besides, we also adapt two representative recommendation architectures [13, 14] into the cold-start setting (CS_NCF and CS_MF). More details about baselines can be found in **Appendix B**.

*Implementation details.* We implement the proposed method with Pytorch framework [21]. Specifically, we integrate two state-of-the-art federated recommendation methods into our framework, named IFedNCF and IPFedRec, respectively. Detailed implementation and parameter configurations are summarized in **Appendix C**.

## 5.3 Comparison Analysis with Baselines (*Q1*)

We compare the model performance with federated baselines and centralized baselines, and then analyze the experimental results.

*Compared with federated cold-start baselines.* As shown in Table 1, we have two observations:

**First, our method consistently performs much better than all federated baselines.** Particularly, FedMVMF, CS_FedNCF, FedVBPR and FedDCN achieve better performance than CS_PFedRec. Recall the optimization procedure of these methods, it utilizes both the user-item interaction information and the item attribute information during model's training phase, indicating that the item

attribute is essential for cold items recommendation. In contrast, CS_PFedRec only takes item attribute as the similarity measure to obtain cold item embedding, performs poorly in cold items recommendation. In our method, the proposed item representation alignment mechanism bridges the connection between attribute representation learned from raw attributes and the item embedding learned from interaction records during optimization. As a result, it facilitates the meta attribute network learning latent item representation that depicts user preferences, and the informative item representation is beneficial for cold item recommendation.

**Second, integrating existing FedRec architectures into our proposed IFedRec framework (IFedNCF and IPFedRec) achieves outstanding performance improvement in all settings.** Our method is a general cold-start FedRec framework, which can be easily instantiated with existing FedRec architectures. Compared with the vanilla FedNCF and PFedRec, our IFedNCF and IPFedRec deploy the meta attribute network on the server side and add an extra item embedding regularization term on the local model's training, which does not change the recommendation model architecture and introduce no extra computational overhead for clients.

*Compared with centralized cold-start baselines.* In addition to the federated baselines, we also conduct experiments to compare our model's performance against centralized baselines. From the Table 2, we can see that **our IFedRec achieves better performance than the centralized baselines on all datasets.** Taking Heater as an example, the performance gain (@20) of our method on the CiteULike dataset are 13.86%, 8.74% and 9.34% on three evaluation metrics, respectively. A similar performance gain trend is also shown in other three datasets. We analyze the reason from two aspects: **First,** for centralized models, all users share the same module parameters in the system. In comparison, our method preserves user embedding and rating prediction modules as personalized

| Methods | | Metrics | CiteULike | | | XING-5000 | | | XING-10000 | | | XING-20000 | | |
|---|---|---|---|---|---|---|---|---|---|---|---|---|---|---|
| | | | @20 | @50 | @100 | @20 | @50 | @100 | @20 | @50 | @100 | @20 | @50 | @100 |
| CenRec | Heater | Recall | 37.17 | 55.13 | 68.52 | 14.51 | 16.09 | 18.16 | 16.60 | 19.48 | 22.48 | 16.94 | 19.74 | 22.55 |
| | | Precision | 8.92 | 5.50 | 3.52 | 5.70 | 2.69 | 1.60 | 8.73 | 4.19 | 2.44 | 8.86 | 4.21 | 2.43 |
| | | NDCG | 31.36 | 35.95 | 37.68 | 8.97 | 7.78 | 7.48 | 14.00 | 12.06 | 11.13 | 13.18 | 11.32 | 10.66 |
| | GAR | Recall | 5.45 | 8.81 | 13.07 | 1.44 | 3.22 | 5.49 | 0.74 | 3.38 | 6.16 | 0.85 | 2.87 | 6.11 |
| | | Precision | 1.42 | 0.91 | 0.66 | 0.69 | 0.55 | 0.46 | 0.37 | 0.67 | 0.62 | 0.45 | 0.51 | 0.39 |
| | | NDCG | 3.43 | 4.42 | 5.48 | 0.89 | 1.57 | 2.32 | 0.80 | 1.86 | 2.87 | 0.85 | 2.02 | 2.97 |
| | CS_NCF | Recall | 29.41 | 46.43 | 61.85 | 18.42 | 32.03 | 45.19 | 21.80 | 35.26 | 47.54 | 19.65 | 33.00 | 45.98 |
| | | Precision | 7.06 | 4.70 | 3.18 | 10.76 | 7.49 | 5.28 | 11.72 | 7.68 | 5.22 | 12.09 | 8.09 | 5.65 |
| | | NDCG | 24.93 | 30.52 | 33.71 | 16.38 | 20.85 | 23.88 | 17.58 | 20.98 | 23.32 | 15.91 | 19.76 | 22.72 |
| | CS_MF | Recall | 1.01 | 2.30 | 4.32 | 0.48 | 1.04 | 1.99 | 0.36 | 0.89 | 1.78 | 0.41 | 0.93 | 1.73 |
| | | Precision | 0.25 | 0.24 | 0.23 | 0.24 | 0.22 | 0.22 | 0.20 | 0.32 | 0.40 | 0.26 | 0.24 | 0.22 |
| | | NDCG | 0.87 | 1.62 | 2.59 | 0.36 | 0.59 | 0.97 | 0.30 | 0.54 | 0.88 | 0.35 | 0.58 | 0.87 |
| Ours | IFedNCF | Recall | **42.32** | **59.92** | **72.89** | **23.48** | **42.05** | **55.45** | **26.97** | **41.57** | **55.37** | **26.36** | **41.44** | **54.48** |
| | | Precision | **9.70** | 5.80 | 3.65 | **13.66** | **9.55** | **6.37** | **14.38** | **9.02** | **6.06** | **16.25** | **10.23** | **6.75** |
| | | NDCG | **34.29** | 37.61 | 38.74 | **20.93** | **27.41** | **29.46** | **21.65** | **24.66** | **27.02** | **21.99** | **25.30** | **27.22** |
| | IPFedRec | Recall | 41.51 | 59.63 | 72.71 | 21.77 | 37.30 | 53.18 | 25.92 | 40.33 | 54.64 | 24.67 | 40.07 | 53.58 |
| | | Precision | 9.48 | **5.81** | **3.67** | 12.75 | 8.76 | 6.12 | 13.84 | 8.77 | 5.97 | 15.29 | 9.92 | 6.66 |
| | | NDCG | 33.48 | **37.69** | **39.07** | 19.74 | 24.77 | 28.34 | 20.66 | 23.90 | 26.52 | 20.53 | 24.49 | 26.91 |

**Table 2: Experimental results of the centralized baselines and our method on four datasets. "CenRec" denotes the centralized baseline sand "Ours" represents that we integrate two state-of-the-art federated models into our framework. Particularly, we report the results in units of 1e-2 and the best results are bold.**

| Methods | CiteULike | | | XING-5000 | | | XING-10000 | | | XING-20000 | | |
|---|---|---|---|---|---|---|---|---|---|---|---|---|
| | Recall | Precision | NDCG | Recall | Precision | NDCG | Recall | Precision | NDCG | Recall | Precision | NDCG |
| IFedNCF | **42.32** | **9.70** | **34.29** | **23.48** | **13.66** | **20.93** | **26.97** | **14.38** | **21.65** | **26.36** | **16.25** | **21.99** |
| w/ LAN | 38.73 | 9.01 | 31.60 | 1.59 | 0.67 | 0.85 | 0.86 | 0.47 | 0.79 | 1.54 | 0.91 | 1.35 |
| w/o ISAM | 0.85 | 0.22 | 0.79 | 0.55 | 0.17 | 0.25 | 0.32 | 0.19 | 0.27 | 0.25 | 0.15 | 0.20 |
| IPFedRec | **41.51** | **9.48** | **33.48** | **21.77** | **12.75** | **19.74** | **25.92** | **13.84** | **20.66** | **24.67** | **15.29** | **20.53** |
| w/ LAN | 38.73 | 8.93 | 31.27 | 2.00 | 0.77 | 0.95 | 0.58 | 0.36 | 0.53 | 0.18 | 0.10 | 0.12 |
| w/o ISAM | 1.05 | 0.26 | 1.03 | 0.27 | 0.15 | 0.21 | 0.42 | 0.24 | 0.38 | 0.46 | 0.27 | 0.39 |

**Table 3: Ablation study for IFedRec on four datasets. "w/ LAN" denotes that we deploy the local attribute network on the client. "w/o IRAM" means to remove the item representation alignment mechanism from our method. We show the results in the units of 1e-2 on @20 metrics.**

components, which is helpful in capturing user preferences and promoting personalized recommendations. **Second,** compared with centralized models, there are more parameters in our method, which enables the system to possess a stronger representation capacity, allowing it to better capture complex patterns and features present in the data and achieve better performance.

## 5.4 Ablation Studies (Q2)

We design model variants to explore the effectiveness of the key modules in our method. For a thorough analysis, we conduct experiments based on IFedNCF and IPFedRec on four datasets and report the results of @20 on three metrics.

***Integrate the attribute network into the local recommendation model.*** To give a more thorough understanding for the cold-start federated recommendation model construction, we build a model variant by deploying the attribute network on each client, named "w/ LAN". Particularly, the local recommendation model replace the item embedding module with the attribute network which takes the item attributes as input. As shown in Table 3, we can see that our method achieves superior performance than the model variant. Compared with it, our IFedRec learns two item representations which enhance the system's ability to identify different items. By effectively learning the item representations, the system can better

understand the inherent item characteristics, such as item similarities and item-item relationships, which in turn lead to more accurate recommendations that align with users' preferences. In addition, our IFedRec maintains the raw item attributes on the server to avert the potential damage from malicious exploitation.

***Remove the item representation alignment mechanism from IFedRec.*** To verify the efficacy of our proposed item representation alignment mechanism for the cold-start recommendation, we construct a variant "w/o IRAM" by removing it from our method. Hence, the learning process of the model is modified as: First, the system optimizes a federated recommendation model whose on-device model is trained with only the recommendation loss. Second, the attribute network on the server is initialized with random parameters and never updated. As in Table 3, removing the item representation alignment mechanism from our method degrades the performance significantly. In our method, by aligning two item representations, the client achieves a more comprehensive item embedding enhanced with attribute representation, which promotes local recommendation model training. Besides, the meta attribute network trained with the item embedding can absorb the user preferences towards items, facilitating the cold item recommendation.

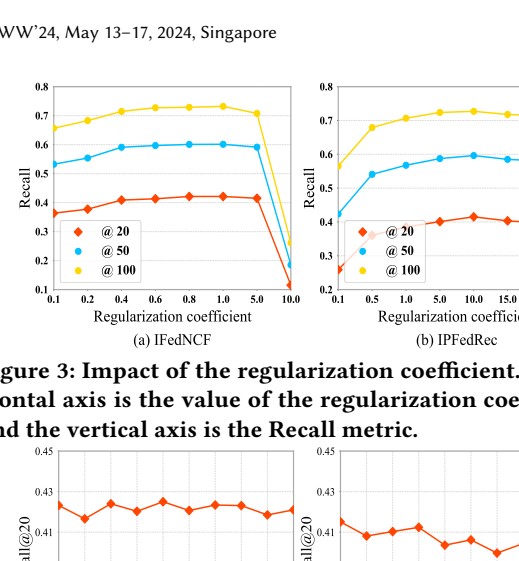

**Figure 3: Impact of the regularization coefficient. The horizontal axis is the value of the regularization coefficient $\lambda$, and the vertical axis is the Recall metric.**

**Figure 4: Impact of the meta attribute network training epoch. The horizontal axis is the value of meta attribute network training epoch $E_1$, and the vertical axis is the Recall@20.**

## 5.5 Impact of Hyper-parameters (Q3)

In this section, we study the impact of two key hyper-parameters of IFedRec: the coefficient $\lambda$ of item attribute representation regularization on the client and the training epochs $E_1$ of the meta attribute network on the server. Particularly, we take the CiteULike dataset as an example and conduct experiments based on IFedNCF and IPFedRec. Due to limited space, we summarize the results in the main text and detailed configurations can be found in **Appendix D**.

**Regularization coefficient $\lambda$.** As shown in Figure 3, we can see that: The performance change trends of IFedNCF and IPFedRec are similar, *i.e.*, as the coefficient increases, the performance first gets better and then decreases. When the regularization coefficient is large, the local recommendation model is injected with too much globally learned item attribute representation information, which interferes with the local model's learning from user preference. As a result, the local item embedding is biased and cannot well characterize user personalization, which leads to a decrease in model performance. The optimal regularization coefficient values for IFedNCF and IPFedRec appear in 1.0 and 10.0, respectively.

**Meta attribute network training epoch $E_1$.** As shown in Figure 4, we find that the performance of IFedNCF is slightly improved as the server training epochs increase. For the IPFedRec, the model gets the best performance when $E_1 = 1$. Hence, one-step optimization is enough to achieve satisfactory performance, which is efficient without much computational overhead.

## 5.6 Convergence with Clients Amount (Q4)

In this section, we investigate the convergence of our proposed IFedRec. Due to limited space, here we summarize the results and conclusions briefly and more details can be found in **Appendix E**. As shown in Figure 5, our method can achieve outstanding performance at a small sampling ratio, *e.g.*, IPFedRec gets 0.4035 on

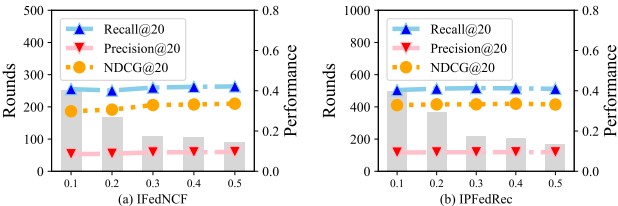

**Figure 5: Convergence analysis about the client amount participated in each communication round. The horizontal axis is the client sampling ratio, and the left vertical axis is the number of communication rounds, the right vertical axis is model performance on three metrics.**

| Methods | Metrics | Noise strength $\delta$ | | | | | |
|---|---|---|---|---|---|---|---|
| | | 0 | 0.1 | 0.2 | 0.3 | 0.4 | 0.5 |
| IFedNCF | Recall | 42.32 | 41.81 | 41.87 | 41.23 | 41.09 | 40.84 |
| | Precision | 9.70 | 9.66 | 9.59 | 9.32 | 9.08 | 8.79 |
| | NDCG | 34.29 | 33.83 | 33.62 | 33.15 | 33.16 | 32.86 |
| IPFedRec | Recall | 41.51 | 41.10 | 40.48 | 40.11 | 40.57 | 39.52 |
| | Precision | 9.48 | 9.49 | 9.31 | 9.49 | 9.45 | 9.03 |
| | NDCG | 33.48 | 33.50 | 33.30 | 32.68 | 32.13 | 31.49 |

**Table 4: Results of privacy-protection IFedRec with various Laplacian noise strength $\lambda$.**

Recall@20, which also outperforms other baselines. On the other hand, more clients participating in a communication round would accelerate model convergence. In summary, IFedRec supports the FedRec system to optimize with insufficient client participation, which is common in physical scenarios.

## 5.7 Privacy-Protection Enhanced IFedRec (Q5)

In this section, we investigate the performance of our IFedRec enhanced with the local Differential Privacy technique. Particularly, we set the Laplacian noise strength from 0.1 to 0.5 with an interval of 0.1 and also conduct the experiment on the CiteULike dataset. We give the experimental results of IFedNCF and IPFedRec of @20 on three metrics. As shown in Table 4, model performance degrades as the noise strength $\delta$ increases, while the performance drop is slight if $\delta$ is not too large. Hence, a moderate noise strength, *e.g.*, 0.2 is desirable to achieve a good trade-off between model performance and privacy protection ability.

## 6 CONCLUSION

In this paper, we introduce IFedRec, the first effort that addresses the new items recommendation scenario in the federated setting. Our two-phase learning framework enables the learning of two item representations to protect private user interaction data while preserving item attributes on the server. The proposed item representation alignment mechanism maintains the correlations between item attributes and user preferences. Then, the cold item could be inferred by the item representations learned by the server. Extensive experiments and in-depth analysis demonstrate the remarkable performance improvement of our model compared to state-of-the-art baselines, particularly in learning cold items. As a general cold-start recommendation framework, IFedRec can be easily combined with existing techniques to explore additional scenarios, such as recommendation diversity and fair recommendation.

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

## A DATESETS

We introduce the datasets below and the detailed statistics are summarized in Table 5.

**CiteULike** is collected from an article recommendation service platform, where the registered users create personal citation libraries recording interested articles. There are $5,551$ users, $16,980$ articles and $204,986$ user-article interactions in the dataset. Each article has a title and abstract, which can be utilized as the auxiliary item information. Following the preprocess procedure of [26, 39], we first calculate the tf-idf to generate an $8,000$ dimension attribute vector for each item, and then utilize SVD to reduce the dimensions to 300. Hence, we obtain a $16,980 \times 300$ item attribute matrix $\mathcal{X}$.

**XING** is collected from the ACM RecSys 2017 Challenge, which has $106,881$ users, $20,519$ items and $4,306,183$ user-item interactions. Each item has a $2,738$-dimensional attribute. Particularly, we conduct three subsets by sampling different user population sizes, *i.e.,* $5,000$, $10,000$ and $20,000$. The items amount of three subset are $18,769$, $20,256$ and $20,510$, and the total interactions are $191,603$, $383,156$ and $768,471$, respectively.

## B BASELINES

We introduce the details about baselines as follows:

- **Heater** [39]: This method first pretrains a collaborative filtering model with user-item rating information to obtain user embedding and item embedding. Then, it trains a recommendation model based on user/item attributes by regularizing the distance between pretrained user/item embedding and learned latent user/item representation.
- **GAR** [7]: This method presents a generative adversarial recommendation model architecture. A generator takes item attributes as input and learns the latent item representation, and a recommender takes the pretrained embeddings as input and predicts rating. The model is optimized by an adversarial loss between the generator and recommender.
- **CS_NCF**: We replace the item embedding module of NCF [13] with a one-layer MLP to learn latent item representation with item attributes and keep other details unchanged. When new items come, each user makes recommendations with their raw attributes based on the trained model.
- **CS_MF**: We first train MF [14] with the warm items. For the cold item, we find the top-k similar warm items by calculating the item-item attribute similarity, and then take the averaged trained top-k warm item embeddings as the cold item representation to make a prediction based on the trained user embeddings.
- **FedMVMF** [10]: This is a matrix factorization method based on multiple data sources, *i.e.,* user-item interaction information and item attribute information. Particularly, each user maintains the user embedding locally and other model parameters are updated on the server.
- **CS_FedNCF**: We adapt the FedNCF [22] into cold-start setting by replacing the item embedding module with a one-layer MLP and keep other details unchanged. The cold item recommendation method is the same as used in CS_NCF.
- **CS_PFedRec**: We first train PFedRec [35] with the warm items. For the cold items, we adopt the same prediction method as in CS_MF.

- **FedVBPR**: VBPR model [12] is a content enhanced recommendation model, which integrates the visual item features into the model to heighten the collaborative filtering framework. We adapt it into the federated learning framework and obtain FedVBPR.
- **FedDCN**: DCN is a deep and cross network architecture, which can capture the complex interactions across multiple item features. We adapt it into the federated learning framework and obtain FedDCN.

## C IMPLEMENTATION DETAILS

For a fair comparison, we set the latent representation dimension as 200 and the mini-batch size as 256 for all methods. For the learning rate hyper-parameter, we tune it via grid search on the validation set. Besides, we resample negative items in each epoch/round and set the sampling ratio as 5 for all methods. For our method, we instantiate IFedRec with two representative FedRec architectures, *i.e.,* FedNCF and PFedRec, and obtain IFedNCF and IPFedRec, respectively. For IFedNCF, we take a two-layer MLP as the rating prediction module. For IPFedRec, we set a one-layer MLP as the rating prediction module following the original paper. On the server side, we deploy a one-layer MLP as the meta attribute network, whose input dimension is the same as the item attribute size and the out dimension is 200. Notably, two centralized baselines Heater and GAR require the pretrained collaborative filtering representations as model input. Hence, we train a matrix factorization model with a latent factor of 200. We report the average results of five repetitions for all experiments.

## D HYPER-PARAMETER ANALYSIS DETAILS

### D.1 Regularization term coefficient $\lambda$

During the training phase, we add the item attribute semantic representation as the regularization term of the local recommendation model by minimizing the distance between it and the local item embedding. According to the validation set performance, we set the regularization term coefficient values on IFedNCF with $\{0.1, 0.2, 0.4, 0.6, 0.8, 1.0, 5.0, 10.0\}$, and on IPFedRec with $\{0.1, 0.5, 1.0, 5.0, 10.0, 15.0, 20.0, 30.0\}$. For conciseness, we only report the results on metric Recall because the patterns on the other two metrics show similar results as Recall.

### D.2 Meta attribute network training epoch $E_1$

On the server side, we deploy a meta attribute network, which takes raw item attributes as input and learns the item latent representation with the global item embedding as supervision. Particularly, we set the Meta attribute network training epochs from 1 to 10 with an interval of 1. For brevity, we report the results on Recall@20.

## E CONVERGENCE WITH CLIENTS AMOUNT DETAILS

We take the CiteULike dataset as an example to conduct experiments. In federated optimization, there is a trade-off between model convergence efficiency and the client's amount of participation in each communication round. Generally, the larger the number of clients sampled in a training round, the faster the federated model

| | Training | | | | Validation | | | Test | | |
|---|---|---|---|---|---|---|---|---|---|---|
| | #users | #items | #interactions | sparsity | #users | #items | #interactions | #users | #items | #interactions |
| CiteULike | 5,551 | 13,584 | 164,210 | 0.22% | 5,551 | 1,018 | 13,037 | 5,551 | 2,378 | 27,739 |
| XING-5000 | 5,000 | 11,261 | 117,608 | 0.21% | 5,000 | 1,878 | 56,465 | 5,000 | 5,630 | 17,530 |
| XING-10000 | 10,000 | 12,153 | 230,765 | 0.19% | 10,000 | 2,027 | 110,731 | 10,000 | 6,076 | 41,660 |
| XING-20000 | 20,000 | 12,306 | 444,199 | 0.18% | 20,000 | 2,051 | 251,735 | 20,000 | 6,153 | 72,537 |

**Table 5: Statistics of four cold-start recommendation datasets. The items are divided into three subsets, where items in the training set are warm items and others are cold items.**

converges. In practical scenarios, due to communication overhead and client computation power limitations, the server usually can only collect a limited number of clients each time to train the model. Especially in the recommendation scenario, the number of clients is large, and it is more difficult to collect enough clients for model training, which poses a challenge for the FedRec system to train the model with limited clients. To this end, we conduct experiments to simulate the setting. Particularly, we constrain the client sampling ratio in each communication round from 0.1 to 0.5 with an interval of 0.1.

