# OpenReview forum: "When Federated Recommendation Meets Cold-Start Problem: Separating Item Attributes and User Interactions"
_ACM.org/TheWebConf/2024/Conference — TheWebConf24 Oral_

### Official Review · Reviewer_EjDW · 2023-11-15

**Novelty:** 7
**Technical Quality:** 6

**Review:**

This paper presents an item-aligned federated recommendation model for new item cold-start recommendation. To be specific, it learns two item representations based on interactions and item attributes and aligns them during the training phase so that the trained model can infer the new items with item attributes. Compared with the centralized model, the proposed method is designed under the FL framework that can preserve the user interaction data in privacy. Besides, this method considers the safety of item attributes by maintaining them on the server from malicious usage, which is rarely explored in the federated recommendation research. Experimental validation shows that the proposed method outperforms both centralized and federated baselines significantly.

Pros:

1.	The paper focuses on the challenging cold-start recommendation task under the privacy constraints, that is, keeping user interactions and item attributes safe, which is an exciting attempt. This brand-new setting poses new challenges to federated learning research and further prompts privacy preservation.

2.	The authors design a novel and intuitive method to utilize cold item attributes for inference, which is easy to follow.

3.	The proposed method achieves outstanding performance gains than many baselines. Also, it seems that the model is robust to system noise.

4.	The paper is well organized and clarifications are clear.

Cons:

1.	In section preliminary, the definition of federated cold-start recommendation does not reflect federated learning well. The recommendation model $\mathcal{F}_\theta$ is learned under FL optimization framework, which should be expressed explicitly to distinguish it from general centralized recommendation models learning.

2.	It is suggested to discuss the potential issues of deploying the proposed model in practical setting. For example, there might have massive new items in the application scenario, how to provide users only with high quality new item representations to local model inference instead of distributing all the items to all users?

**Questions:**

1.	In figure 5, what is the meaning of the gray column? Is it the total communication rounds number that is set for different client amount models’ training? If so, how to determine the appropriate values?

2.	In addition to the new items recommendation scenario, can the proposed method generalize to the various general recommendation settings, e.g., warm items recommendation and sequential recommendation?

**Reviewer Confidence:**

4: The reviewer is certain that the evaluation is correct and very familiar with the relevant literature

**Scope:**

4: The work is relevant to the Web and to the track, and is of broad interest to the community

---

### Official Review · Reviewer_k2bJ · 2023-11-15

**Novelty:** 6
**Technical Quality:** 7

**Review:**

This paper proposes IFedRec, a federated recommendation algorithm that enhances the existing solutions by learning the item representation with attribute information on the server, which can solve the cold-start task. The authors conduct experiments on four benchmark datasets, and the results show that the proposed method can achieve the best performance in terms of three metrics. Further experiments in ablation studies show the effectiveness of the two components in the method.

Strength:
1). The studied issue is important. Privacy-protection based cold-start recommendation is meaningful and deserves to be well considered in recommender system, which matches the conference scope.

2). The efficacy and compatibility are well supported by the experimental evaluation. Results show that the proposed method can remarkably enhance existing methods. Also, the authors present detailed empirical analysis, including the privacy-protection scenario with local Differential Privacy technique. It is technically sound while defeating diverse baselines on benchmark datasets.

3). The authors provide code implementation for performance verification, which is noteworthy and instrumental for experiment reproducibility.

Weakness:
1). The scalability of the proposed method in handing large-scale datasets was not extensively discussed.

2). There are some typos. For example, in the table 3, the item representation alignment mechanism is named IRAM in table caption but written as ISAM in table.

**Questions:**

1). Why the performance of baseline CS_FedNCF and ablation model IFedRec w/ LAN is quite different? Two models share the same model architecture and deploy the attribute network locally. Are there any implementation differences between two models?

2). For the privacy-protection enhanced model, the authors add noise to the item embedding, why not add to other modules?

**Reviewer Confidence:**

4: The reviewer is certain that the evaluation is correct and very familiar with the relevant literature

**Scope:**

4: The work is relevant to the Web and to the track, and is of broad interest to the community

---

### Official Review · Reviewer_ceGE · 2023-11-23

**Novelty:** 6
**Technical Quality:** 5

**Review:**

This paper devises a two-phase framework to learn item attribute – item representation mapping with warm items and infer on the cold items. According to the extensive experiment results, this method is effective to solve the federated recommendation cold-start problem.
Strength:
a. This paper addresses the cold-start problem in federated recommendation, which deserves research attention.
b. The authors conduct thorough experiments which shows the validity of the proposed method, including framework compatibility, module contribution, and convergence analysis. Various state-of-the-art baselines are defeated.
c. The proposed framework shows significant effectiveness on cold start problem, e.g., around 10% performance improvement on CiteULike dataset even over the centralized baseline Heater on all metrics.
Weakness:
a. Related works lack proper citations for collaborative filtering, content-based, and hybrid models, which is less informative.
b. Item representation and item embedding are misunderstanding to some extent. They are both learnable, but derive from different source. It’s better to rename the item representation to alleviate confusion.
c. It deserves consideration to introduce the notation system clearly. Especially in Figure 2, you can add notations along with the description.

**Questions:**

1. In Section Introduction paragraph 2, the authors mentioned federated recommendation system guarantees privacy since users preserve private data locally. Does that mean the users’ privacy is guaranteed but the server’s privacy (item attributes) is leaked? If so, the authors should check the full paper for this misunderstanding description.
2. How is the training and inference efficiency of the proposed framework?

**Reviewer Confidence:**

4: The reviewer is certain that the evaluation is correct and very familiar with the relevant literature

**Scope:**

4: The work is relevant to the Web and to the track, and is of broad interest to the community

---

### Decision · Program_Chairs · 2024-01-22

**Decision:**

Accept (Oral)

**Comment:**

This paper proposes a two-phase approach to map item attribute-item representations using warm items and then apply this to cold items. The reviewers largely agree on the innovative nature and excellent quality of the research. Furthermore, the authors have successfully responded to the reviewers' questions in a satisfactory manner.